# Microplastics in Freshwater: A Focus on the Russian Inland Waters

Yulia Frank [1,*], Alexandra Ershova [1,2], Svetlana Batasheva [3], Egor Vorobiev [1], Svetlana Rakhmatullina [1], Danil Vorobiev [1] and Rawil Fakhrullin [1,3]

[1] Research Center "Microplastic Siberia", Tomsk State University, Lenina av., 36, 634050 Tomsk, Russia
[2] PlasticLab, Russian State Hydrometeorological University, Voronezhskaya Str., 79, 192007 Saint-Petersburg, Russia
[3] Institute of Fundamental Medicine and Biology, Kazan Federal University, Kreml uramı 18, 420008 Kazan, Republic of Tatarstan, Russia
* Correspondence: yulia.a.frank@gmail.com

**Abstract:** The low production costs and useful properties of synthetic polymers have led to their ubiquitous use, from food packaging and household products to high-tech applications in medicine and electronics. Incomplete recycling of plastic materials results in an accumulation of plastic waste, which slowly degrades to produce tiny plastic particles, commonly known as "microplastics" (MPs). MPs can enter water bodies, but only recently the problem of MP pollution of sea and fresh waters has become clearly evident and received considerable attention. This paper critically reviews the accumulated data about the distribution of MPs in the freshwater ecosystems of Russia. The available data on MP abundance in the lakes and river systems of the Russian Federation are analyzed (including the large Lakes Baikal, Ladoga, Onego, Imandra and Teletskoe, and the Volga, Northern Dvina, Ob, and Yenisei Rivers within their tributaries) and compared with the data on freshwater MP contents in other countries. In Russia, the main sources of MP pollution for rivers and lakes are domestic wastewater, containing microfibers of synthetic textiles, fishing tackle, and plastic waste left on shores. Among the MPs detected in the surface waters and bottom sediments, polyethylene (PE), polypropylene (PP), and polyethylene terephthalate (PET) particles predominate. The most common types of MPs in the surface freshwaters are fibers and fragments, with fibers prevailing in the bottom sediments. The reported average MP concentrations in the waters range from 0.007 items/m$^3$ at the mouth of the Northern Dvina River to 11,000 items/m$^3$ in the Altai lakes. However, the estimates obtained in different studies must be compared with great precaution because of significant differences in the methods used for MP quantification. The approaches to further improve the relevance of research into MP pollution of fresh waters are suggested.

**Keywords:** microplastics; freshwater ecosystems; rivers; lakes; bottom sediments; plastic pollution

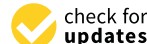



## 1. Introduction

The invention of the first phenol-formaldehyde-based synthetic plastic Bakelite dates back to 1907 [1]. The use of plastic materials began to increase significantly from the middle of the 20th century, when new types of polymers were synthesized. Whereas in 1950, 1.5 million tons of synthetic polymers were produced annually, in 2020 the global production amounted to 367 million tons [2]. Around 99% of plastics are produced from hydrocarbon raw materials [3]. Currently, the use of plastic materials is extremely versatile due their properties, such as low density, low thermal and electrical conductivity, and corrosion resistance. The low costs of production also contribute to their ubiquitous use, from food packaging and household products to high-tech applications in medicine and electronics [4]. Thousands of different polymers are produced today on an industrial scale. The largest shares of the total volume produced currently belong to polyethylene (PE) at

30.3%, polypropylene (PP) at 19.7%, polyvinyl chloride (PVC) at 9.6%, and polyethylene terephthalate (PET) at 8.4% [2].

The increasing production and consumption of plastic materials have gradually become a huge environmental problem due to the widespread pollution of marine and freshwater ecosystems [5–8]. Today, enough data have been accumulated on the direct impact of plastics on hydrobionts, which are associated with the ingestion, suffocation, entanglement, and other mechanical effects, so plastic waste has been recognized as dangerous for animals [9]. However, this is not the only adverse effect of plastic accumulation in the oceans and fresh waters. Natural conditions in aquatic ecosystems, such as currents, wave dynamics, solar radiation, and aquatic microorganisms, cause slow degradation and fragmentation of plastic objects into smaller particles commonly known as "microplastics" [10]. Microplastics (MPs) also enter the water directly in the form of tiny particles of polymeric materials used in industry and households [11]. The interest of researchers and the number of works devoted to the sources, distribution, circulation, and bioaccumulation of MPs in aquatic environments have increased dramatically after the publication of an article by Thompson et al. (2004) [12], which showed the widespread distribution and accumulation of plastic microfragments and microfibers in the oceans.

MPs are a heterogeneous type of pollutants with a wide range of properties, such as polymer type, density, size, and shape [13]. The diverse characteristics make MPs potentially accessible to a wide range of neuston, pelagic, and benthic species. These pollutants are present in a variety of ecological niches and are able to enter aquatic food webs at different trophic levels [14]. Polymer microparticles travel long distances and can interact with various hydrobionts, from microorganisms [15] to fish and large mammals [16,17].

The widespread pollution of the world's oceans by MPs has become a serious problem. MPs have been found in the water column and bottom sediments of all seas and oceans [12,18,19]. There are five known "garbage patches" in the Atlantic and Pacific Oceans, where plastic debris accumulates in subtropical gyres. Particular attention is now focused on the Arctic region, where the sixth garbage patch is being formed in the Barents Sea [20,21]. The Arctic Ocean has been shown to be contaminated with over 300 billion MP particles [20]. At the same time, recent data show that there is a contrast between the MP content in the surface waters of Atlantic origin and in the waters of river plumes, where the MP content is lower [22].

The studies on MPs in the Russian Federation surface waters also focus chiefly on the seas and are mainly represented by quantitative assessments. Marine plastic pollution has been confirmed by field studies in 7 out of 12 seas studied [23]. The most studied is the Baltic Sea region, where the investigations of MP pollution in the aquatic environment and along the coasts are directed at studying the peculiarities of the distribution and behavior (settling, etc.) of MP particles in the water column [24–26], developing methods for monitoring plastic pollution, and studying the mechanisms of accumulation of marine litter on the coasts of the southeastern Baltic [27–29] and the eastern part of the Gulf of Finland [30–32]. It was found that the maximum amount of MPs is observed in the easternmost part of the Baltic—in the Gulf of Finland—because of the influence of the largest megalopolis of Europe, St. Petersburg, and the special hydrodynamic regime of the estuary of the Neva River, which is a man-made lagoon, where most of the MPs carried by the flow of the Neva River are deposited [33].

Quantitative estimates of the MP content in the surface and subsurface water layers of the seas of the Arctic region have recently been published [22,34,35]. These studies showed the maximum content of marine litter on the coasts and of MPs in the surface water layers in the seas of the Western Arctic—the White, Barents, and Kara Seas. The results confirmed the theory of Van Sebille et al., [21] about the accumulation of pollutants off the coast of Novaya Zemlya, where MP concentrations were an order of magnitude higher than their concentrations in other parts of the Arctic and were comparable to the values obtained in subtropical centers [36].

In contrast to the studies of marine plastic pollution, the projects on the quantification and ecotoxicology of MPs in continental freshwater ecosystems in the Russian Federation are still very few, especially given the vast freshwater resources in this area. Moreover, compared to the data accumulated for the freshwater bodies in European territories, the USA, Southeast Asia, and other regions of the world, the fresh waters in the Russian Federation remain significantly understudied. The purpose of this paper is to systematize and critically review the accumulated knowledge about the distribution of MPs in the freshwater ecosystems of the Russian Federation. To achieve this goal, an analysis of modern literary and information sources devoted to the problem of MP pollution of aquatic ecosystems was conducted with a focus on the freshwater bodies in the Russian Federation.

## 2. Microplastics as Pollutants of Aquatic Ecosystems: State of Research for Freshwater Bodies in the Russian Federation

### 2.1. Overview of the Sources, Sampling, and Analysis of Microplastics in Aquatic Ecosystems

The term "microplastics" was first used in 2004 by Thompson et al. [12] to describe microscopic plastic particles that accumulated in the water column and bottom sediments of the UK coastal aquatic ecosystems. Later, it was proposed to classify all plastic particles smaller than 5 mm as MPs [37]. The Group of Experts on the Scientific Aspects of Marine Environmental Protection [38] expanded the concept by defining it as "plastic particles less than 5 mm in diameter, including nano-sized particles (down to 1 nm)" [39,40].

Some authors have proposed to consider particles smaller than 1 mm along the longest axis as MPs and to classify particles larger than 1 mm as mesoplastics (up to 25–100 mm) [41]. One of the suggested definitions of MPs is as follows: "Microplastics are any synthetic solid particle or polymeric matrix, with regular or irregular shape and with size ranging from 1 μm to 5 mm, of either primary or secondary manufacturing origin, which are insoluble in water" [12]. The U.S. National Oceanic and Atmospheric Administration (NOAA website), which is actively working towards developing a methodology for studying the distribution of MPs in aquatic environments, and other authors [8,42] also support the current definition of MPs as polymer particles less than 5 mm along the longest axis. There are special terms for fine fractions of MPs, such as "nanoplastics" for particles from 1 nm to 1 μm [43] and "mini-microplastic" for particles less than 330 μm [44]. Small MP particles (up to 0.45 mm, including nanoplastics) are also referred to by the general term "submicroplastic" [45].

Small particles are formed in the aquatic environment in the course of the sequential decomposition of larger plastic materials, mainly as a result of the action of physical and chemical factors [10]; these are the so-called "secondary MPs". Plastics can also enter water systems directly in the micro-sized (<5 mm) form [11]. This group of MPs is referred to as "primary MPs".

MPs enter freshwater ecosystems from various point and diffuse sources [46]. Diffuse sources (for example, plastic waste coming with watercourses from a catchment area, with groundwater) are distributed over large areas, while point sources combine direct entry with wastewater, including sewage, agricultural wastewater, storm water, industrial wastewater and others. Primary MPs enter aquatic ecosystems in the form of granules used in many industrial processes (raw materials for the production of plastic products, industrial abrasives, components of paint coatings, drilling fluids, etc.) and in personal care products [42]. The amount of primary MPs entering the oceans annually is estimated at 0.8–2.5 million tons [11]. Secondary MPs are also widespread in aquatic environments. They are formed from the fragmentation of larger plastic products, including plastic waste, synthetic textiles, etc. Quantitative estimates suggest that between 4.8 and 12.7 million tons of plastics enter water bodies annually due to poor waste management [47]. The rates of secondary MP formation are not clearly determined, since this process is complex and depends both on the properties of the material itself and on environmental factors. The leading factor in the degradation of plastics is photochemical oxidation under the action of ultraviolet radiation from sunlight [4,48]. Photooxidation occurs most rapidly on beaches and in open ground conditions; in water, this process is greatly slowed down due to lower

ambient temperatures. The formation of biofilms on plastic surfaces also significantly (up to 99%) reduces the effect of ultraviolet radiation [49]. Other significant environmental factors affecting the decay of large fragments to MPs include wind action, waves, aggressive chemical environments, and microbial degradation processes [50]. The cumulative effect of these factors on the formation of secondary MPs has not been studied enough. However, it is known that the smaller the fragments, the faster their further decomposition is, with the formation of MPs under the influence of UV radiation and mechanical abrasion for all types of polymers [51]. High temperatures accelerate the processes of plastic degradation, in particular its photodegradation [52].

Washing of synthetic textiles has recently been recognized as one of the largest sources of microfibers [53]. It is estimated that millions of fibers enter the wastewater during a typical home clothing wash [54]. Ross et al. [55] revealed the intense pollution of the Arctic waters with synthetic fibers, mainly polyester particles, coming with ocean currents and with atmospheric flows from the south. The study [56] showed that it is microfibers that represent the major part of MP particles in the surface layers of the Arctic waters, as a result of the breakdown of larger synthetic materials. These are shipping waste (primarily discharges of liquid household waste), fishing waste (pieces of plastic nets), as well as sewage brought by currents and waste from offshore platforms. This conclusion was confirmed by a recent study in the Arctic seas adjacent to the Russian Federation [34,57], which showed that most of the particles found in the surface water layer are microfibers of polymers, such as polyethylene terephthalate (PET), polypropylene (PP), and polyethylene (PE).

In general, street runoff, wastewater treatment plants, and atmospheric transfer from land are cited as the largest channels for MPs' entry into aquatic ecosystems [11]. Many authors have confirmed that MP concentrations are often elevated near point sources, such as large population centers, wastewater treatment plants, landfills, and plastic manufactories, and they decrease with distance from the sources [58–60]. Considerable attention is paid to assessing the contribution of treatment facilities to the pollution of water environment with MPs. Although most wastewater treatment plants generally have relatively high MP removal rates (over 95%) [61,62], many wastewater treatment plants are not effective at capturing certain types of particles characterized by small size and high buoyancy, such as microspheres and microfibers [63,64]. Industrial sources can also cause the entry of primary MPs into surface waters. In addition to municipal wastewater treatment plants, high concentrations of MPs in the aquatic environment are observed when sampling in the immediate vicinity of plastic factories and other industrial enterprises. MPs are used in various industrial processes as raw materials ("primary granules") and as a part of abrasive products, and they enter watercourses with regular or accidental releases [65]. Point sources of MPs are characterized by specific "profiles" that reflect the nature of pollution. Microspheres used in personal care and cosmetic products, along with synthetic textile microfibers, are most abundant near wastewater outlets [63,64]; high concentrations of polyester fibers have also been observed near textile factories [66]; microgranules are typical for sites located near the production of plastic products [67]; and fragments of composite thermoplastics containing reflective glass spheres can be associated with the entry of road marking components into surface waters with storm runoff [68]. Most MPs entering the seas from rivers are represented by particles of synthetic polymers left after incomplete wastewater treatment (42%) and microfibers of synthetic fabrics (29%), followed by fragments and fibers from the breakdown of plastic waste (19%) and microspheres from personal care products and industrial sources (10%) [69].

Both special and improvised means can be used to take samples of surface waters and bottom sediments for the quantitative analysis of MPs. Trawl nets of various modifications, such as Manta trawl, neuston and plankton nets, and pump filtration, are used to sample MPs from water, while bottom sediments are collected using bottom grabs or hand tools [32,70–72]. Then, laboratory processing of the samples is performed to selectively extract MPs and get rid of other particles, such as minerals and organics. The identification of MP particles becomes

more difficult in the presence of organic residues, which also have a low density and are not removed during the separation of particles in saturated salt solutions. Therefore, the elimination of impurities of biological origin, which is achieved by using acidic, alkaline, and enzymatic hydrolysis or strong oxidizing agents, is of great importance for adequate quantitative and qualitative analyses of plastic particles [73].

It is now accepted that visual analysis using stereomicroscopy can be used to obtain preliminary quantitative data on the presence of MPs in aquatic ecosystems. For the qualitative identification of plastic particles, spectroscopic methods are used, such as Raman and IR-Fourier spectroscopy, and pyrolysis gas chromatography/mass spectrometry (pyrolysis GC-MS). The chemical composition of particles is determined to understand the ratio of polymer types in the samples and obtain more accurate estimates of the origin and sources of MPs [70]. For particle analysis, some researchers also use scanning electron microscopy combined with energy-dispersive spectroscopic analysis (SEM-EDS) (for example, [74,75]. According to some estimates, the imperfection of the sampling methodology leads to an underestimation of the real concentrations of MPs in water and bottom sediments of freshwater ecosystems [76]. The use of different methods by different research groups for sampling natural waters and sediments and their subsequent laboratory analysis for MP content in many cases make it difficult to compare quantitative data.

### 2.2. Abundance and Distribution of Microplastics in Freshwater Ecosystems

The accumulation and transport of plastic debris and microparticles in continental freshwater ecosystems has become the subject of systematic research only since 2010 [58,77–79]. Terrestrial fresh waters need intensive research on the extent and peculiarities of pollution with MPs, as they represent the most important source of plastic pollution to the oceans [76,80]. The knowledge of the distribution of MPs in freshwater systems remains limited, but it is clear that the amount of pollutants carried by rivers is enormous. The global model of plastic transport to the World Ocean from river flows estimated the total flux at 1.15 to 2.41 million tons [81], with the 20 most polluted world rivers carrying about 67% of the total volume of transported artificial polymers. Quantitative estimates had been performed for large European rivers. The modeling of MP transport in the Danube basin and the calculated data based on the actual concentrations in the surface waters indicate an annual transport of 500 to 1534 tons of MPs into the Black Sea [69,82,83]. For the Po and Rhine Rivers, the calculated values of MP transport to the seas are 120–399 tons and 20–105 tons per year, respectively [69,83]. The Neva River is presumably one of the main sources of MPs entering the waters of the Gulf of Finland [33], while the Northern Dvina is a source of pollution for the White and Barents Seas [34,36,84].

The second reason for the intensive research on freshwater ecosystems, especially rivers, is that they are valuable water resources subjected to pollution [68]. Despite the common belief that most land-based plastic is transported directly to oceans by rivers, available evidence suggests that the bulk of the pollutants accumulates in rivers and their floodplains [85–88]. It is believed that lakes act as filters, retaining MPs in land surface waters as they flow into the continental seas and the World Ocean, and become the primary MP reservoir [89]. As a river flows into a lake, MPs are carried by the surface currents of the lake and can be concentrated in small temporary whirlpools [90]. Wind driven surface currents, especially during storms, also transport and deposit significant amounts of plastics on lake shorelines [91]. Rivers and lakes act not only as transit systems on their way to the ocean, but also as reservoirs of plastic pollution. River estuaries function in the same way: for example, the estuary of the Neva River, at its confluence with the Gulf of Finland of the Baltic Sea, serves as an accumulator of marine debris and MPs in the Neva Bay, where the maximum concentrations of MPs in the water and on the coasts are observed [33].

Regression analysis of the data from several dozen freshwater bodies around the world showed that the part of the world with the highest MP content is Asia, followed (in descending order) by North America, Africa, Oceania, South America, and Europe [92].

The average MP concentrations in the surface waters of freshwater bodies around the world vary from tenths to hundreds and thousands of items per cubic meter. However, quantitative estimates are highly dependent on the methods used for sampling and MP detection. For example, when sampling with a neuston network with a mesh diameter of 0.33 mm in the tributaries of the Great Lakes in the United States and visualizing particles under a binocular microscope, the researchers registered the presence of 4.2 MP items per cubic meter on average [93]. Similar results of 5.60 and 5.57 items/m$^3$ were obtained for the surface waters of the Rhine River (from Basel to Rotterdam) [94] and Elba River [95], respectively, using a 0.30 mm and a 0.15 mm mesh and visual analysis with selective particle identification by (Fourier-transform infrared spectroscopy) FT-IR spectroscopy. The average concentration of MP particles in the water of the Danube River discovered during multiple sampling using a 0.50 mm mesh and subsequent visual analysis was an order of magnitude lower, 0.32 items/m$^3$ [82]. On the contrary, much higher average values (more than 100 items/m$^3$) were detected using a light microscopy in the surface waters of a Seine tributary, the Marne River, when sampling with a Manta trawl with a mesh of 0.08 mm [96]. One of the highest average MP concentrations in river waters, registered by a visual method in samples taken from the Los Angeles River (USA) using a Manta trawl and various types of nets, reached 3473.42 items/m$^3$ in some seasons [77]. However, it cannot be concluded that a finer mesh diameter when filtering water for MP collection is unequivocally associated with an increase in the number of plastic particles. Thus, in the waters of lakes in China, after direct filtration of water samples using a 45 $\mu$m sieve and subsequent visual analysis with verification using a Raman spectroscopy, an average of 617 to 7050 particles per cubic meter was recorded [97,98]. Hundreds and thousands of MP particles per cubic meter of water were detected in the Nakdong River in South Korea using 20 $\mu$m filters [99], and the volume of MPs carried by the river per year was estimated by the authors at 5.4–11 trillion particles or 53.3–118 tons.

MPs are also ubiquitous in freshwater bottom sediments where their concentrations vary from single and tens of particles per kg dry weight [100,101] to thousands; for example, in the sediments of the Amsterdam canals, the MP content was found to reach 10,500 particles per kg dry weight [102]. Bottom sediments are considered to be a MP depot, the accumulation of particles in which is the result of sedimentation on a long-term scale.

In addition to the difficulties associated with comparing data obtained by different methods, surface water bodies have other features making it difficult to quantify the MP content. The processes of transport and redistribution of MPs in surface water bodies, especially in river systems, are complex. The mobilization, transport, dispersion, and accumulation of plastics in rivers depend on hydrological and meteorological conditions, including wind speed and direction [103,104], flow velocity, and water level and discharge [105,106]. The hydromorphological and dynamic features of surface water bodies determine the zones of accumulation and the surface transport of MPs [80].

The ways of MP transport in aquatic ecosystems are complex. They are currently best understood for marine and brackish waters and include surface drift, distribution in the water column due to vertical mixing, aggregation near shores and natural obstacles, and sedimentation [107]. In addition to the dynamic conditions of aquatic environments, MP distribution is affected by their physical characteristics (shape, particle size, and polymer density), and, as a result, particles in aquatic environments demonstrate a high variability of dynamic properties, including settling/rising velocity, critical shear stress, and re-suspension threshold [108,109].

To a large extent, the fate of MPs in marine and brackish water ecosystems depends on the type of polymer. Typically, PP and PE are low-density plastics that are relatively buoyant and are carried by currents, while PVC, PS, polyester, and polyamides are considered high-density plastics that tend to sink [110,111]. However, even PP and PE can acquire a higher density as a result of the addition of mineral additives [112]. The denser varieties of plastics tend to sink and reach the bottom sediments. A significant number of MPs are eventually buried in deep ocean sites [113] and accumulate in food chains [114].

The hydrodynamics influencing MP behavior in freshwaters is relatively poorly studied in comparison to that in the waters of the seas. However, it is known that, since the specific gravity of plastics primarily affects their distribution in the surface waters, the water column, and the bottom sediments, MPs are distributed in freshwater systems along a gradient from top downward, with a pronounced increase in concentrations from the surface to the bottom [79,107]). The difference between the MP content in fresh surface waters and bottom sediments can be enormous. Thus, MP concentrations in the sediments of the Elbe River in Germany are, on average, 600,000 times higher than those in the water [95].

Due to the peculiarities of the transport and the accumulation of MPs in rivers, it is difficult to interpret the data when studying MP distribution in cases where it is impossible to take samples along the transect. The selection and analysis of point samples presents a mosaic picture and does not always reflect the real MP concentrations. Fluctuations in the distribution of MPs in the water along the river are due to the fact that complete mixing and redistribution of pollutants can occur at a considerable distance from the source or confluence site with another watercourse; currents, turbulence, and wind action can contribute to the accumulation of floating particles in bends, and the slowing down of currents may be associated with biofouling and the sinking of denser fragments to the bottom [79]. In addition, a set of point measurements is only a "snapshot", which makes it difficult to estimate the total particle flux, and therefore, for surface water bodies, it is preferable to conduct spatiotemporal studies to determine the average MP concentrations over a certain representative period of time [83].

Despite the fact that the distribution and the abundance of MPs in freshwater ecosystems are influenced by many factors, in general, MP concentrations in freshwater ecosystems, especially in rivers, are directly dependent on population size and population density, but are also influenced by such factors as the efficiency of wastewater treatment, volumes of discharged wastewater, and remoteness from urbanized, industrial, and agricultural centers [76,115]. The majority of MP particles entering freshwater ecosystems are secondary in origin. They form as a result of the destruction of larger plastic items, such as single-use packaging, tires, and fibers, and they are also represented by particles of paving and car paints [68]. These types of MPs enter water bodies and streams along with surface and agricultural runoff or directly from plastic waste as a result of inefficient waste management [116]. Storm runoff is another major source of MPs entering freshwater bodies. These types of wastewater introduce to the aquatic environment the particles from car tire abrasion and road markings [68,117]. However, high MP concentrations in river waters are often associated with primary MPs. The plastics in the Austrian Danube were mainly in the form of industrial raw materials, such as pellets and flakes [82]. Two studies showed that most of the plastics found in surface waters came from cosmetics or textiles [118,119]. In a tributary of the Ob River, the Tom River, synthetic microspheres were identified, the share of which reached 56.8% of the total amount of particles found downstream of a large industrial center of Western Siberia, the city of Kemerovo [120].

*2.3. Detection and Quantification of Microplastic Content in Freshwater Lakes and Rivers in the Russian Federation*

The freshwater pool of the Russian Federation is huge, ranking second in freshwater reserves after Brazil in the list of nine countries containing 60% of the world's freshwater resources [121]. Figure 1 shows a fragment of the map of litter distribution (mainly plastics at 61.3%) in surface waters, built as part of the LITTERBASE project [122], based on an analysis of 1359 publications. To date, the database has deposited data on microparticles less than 5 mm along the maximum axis for only two freshwater river systems—the Ob River [120] and the Yenisei River [123]. We have supplemented the map with new data on the detection of MPs in fresh waters in the Russian Federation (Figure 1). Tables 1 and 2 present the details of published studies on the content of MPs in the waters and bottom sediments of freshwater bodies belonging to different watersheds.

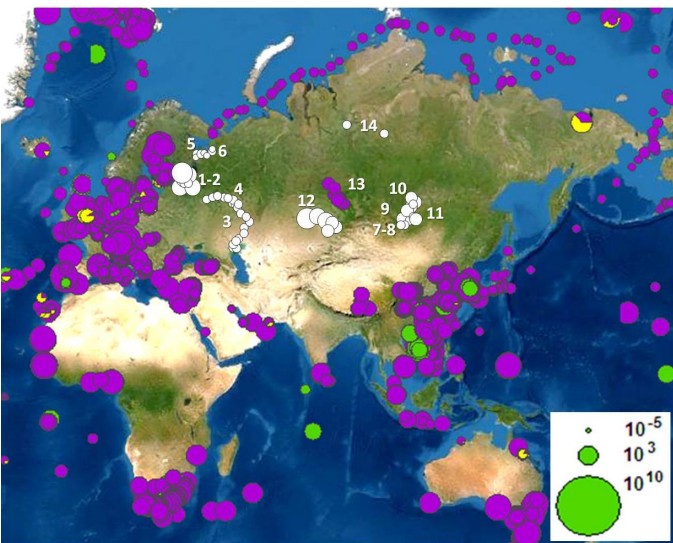

**Figure 1.** Published data on the quantitative assessment of marine and freshwater litters (the MPs included are shown by violet circles; the diameter of the circles reflects the particle concentrations) according to the LITTERBASE [122] and other data on MP abundance in the Russian Federation inland waters (white circles). Designations: 1—Ladoga Lake with tributaries [124], 2—the Smolenka and Neva rivers [125], 3—the Volga River [126], 4—the Volga River system [127], 5—the Northern Dvina River [84], 6—freshwater lakes and rivers in the White Sea basin [34], 7—Lake Baikal [128], 8—Lake Baikal [129], 9—Lake Baikal [130], 10—Lake Baikal [131], 11—Lake Baikal [132], 12—freshwater lakes in Altai [75], 13—the Ob River system [120], and 14—the Yenisei River system [123].

2.3.1. Microplastics in the Freshwater Bodies of the European Part of the Russian Federation

One of the early papers was devoted to determining the content of MPs in the water and the surface layer of bottom sediments of Lake Ladoga and its tributaries [124]. This preliminary study revealed relatively high concentrations of particles in the surface waters of the lake and the rivers flowing into it—the Neva, Almoga, Morie, Vuoksa, Burnoy, and Volkhov (Table 1, Figure 1). The concentration of MP particles in the bottom sediments of Lake Ladoga exceeded the values in the water by 100 times on average (Table 2). The results obtained indicate the important role of bottom sediments as an active zone of MP accumulation and deposition. In the next work of the authors, the concentration and the chemical composition of MP particles in the water, coastal soils, and bottom sediments of the Neva Bay of the Gulf of Finland were analyzed, including the mouth sections of the Neva itself and the small rivers Smolenka and Malaya Neva, which flow through the territory of the St. Petersburg agglomeration [125]. After visual sorting, the chemical composition of the polymers was determined using a combination of FT-IR and Raman spectroscopy. In the surface waters of the mouths of the Smolenka and Neva Rivers, 1.10–3.00 items of MPs per 1 L, which meant 1100–3000 particles per cubic meter, was found (Table 1). The lower size limit of the analyzed particles was 0.1 mm; for water samples, it was determined based on the diameter of the mesh of the filtering device. The bottom sediments in the estuarine areas of the rivers flowing into the Neva Bay were also examined for the content of MPs. It was found that the content of plastic particles varied from 30.0 items/kg dry weight in the bottom sediments of the Malaya Neva River to 120 items/kg in the Neva sediments. In the samples of water and bottom sediments in the mouths of the rivers, microfibers predominated (more than 95% of the total MPs), with particles <1.50 mm along the longest axis being the most common. The dominant polymer type was PET. After comparing the concentrations of MPs in the mouths of the inflowing rivers and other parts of the bay, the authors concluded that the Neva River flow is the main source of water pollution in the Neva Bay.

**Table 1.** Published data on the abundance of MPs in the surface waters of Russian rivers and lakes (data are presented with increasing concentration).

| Water Body | Drainage Basin | Character of the Study | Sampling Method/Determination of the Water Volume | Volume per Sample, L | Detection and/or Identification Methods | Predominant MP Shape/Polymer Types | Average MP Counts, Items/m³ | Reference |
|---|---|---|---|---|---|---|---|---|
| Northern Dvina River, mouth | Arctic | S/T | neuston net, 0.33 mm/C | nd | visual, FT-IR | fragment/PE, PP | 0.007 | [84] |
| Baikal Lake | Arctic | S | drift net, 0.30 mm/C | 352,000–494,000 | visual, FT-IR | film/PE, PP | 0.27 | [130] |
| Freshwater lakes (White Sea basin) | Arctic | S | direct sampling, 0.10 mm filter/DM | 500 | visual, FT-IR | nd/PE | 0.50 | [34] |
| Volga River | internal watershed | S | manta trawl, 0.30 mm/IM | 25,000–130,000 | visual, DSC | fragment/PE, PP | 0.90 | [126] |
| Baikal Lake, southern part | Arctic | S | tugging neuston net, nd/nd | nd | visual | fiber/nd | 1.07 | [128] |
| Kedovka, Koida, Onego, Vaga rivers (White Sea basin) | Arctic | S | direct sampling, 0.10 mm filter/DM | 500 | visual, FT-IR | film, fiber/PE, PVC | 1.67 | [34] |
| Baikal Lake, southern part | Arctic | S | direct sampling, GF/F filter/DM | 1.50 | visual | fiber, fragment/nd | 1.79 | [132] |
| Yenisei River | Arctic | S | manta trawl, 0.33 mm/IM | 8610–12,400 | visual, hot needle | fiber/nd | 2.89–3.01 | [123] |
| Nizhnaya Tunguska River (Yenisei River system) | Arctic | S | manta trawl, 0.33 mm/IM | 13,100–29,200 | visual, hot needle | fiber/nd | 1.20–4.53 | [123] |
| Baikal Lake, southern part | Arctic | S | neuston net, nd/nd | 16,000–52,000 | visual | fiber/nd | 0.51–6.46 | [129] |
| Tom River (Ob River system) | Arctic | S | manta trawl, 0.33 mm/C | 50,100–61,400 | visual, hot needle | fragment/nd | 44.2 | [120] |
| Ob River | Arctic | S | manta trawl, 0.33 mm/C | 17,700–62,500 | visual, hot needle | fragment/nd | 51.2 | [120] |
| Baikal Lake | Arctic | | pumping, plankton net, 0.02 mm/DM | 300 | visual, FT-IR microspectroscopy | fragment/PP, PET | 291 | [131] |
| Smolenka River, mouth | Atlantic | S | filtering device, 0.10 mm/DM | nd | visual, FT-IR, Raman microspectroscopy | fiber/PET | 1100 | [125] |
| Urban water bodies in Nizhny Novgorod (Volga River system) | internal watershed | S/T | direct sampling, 0.13 mm filter/DM | 10.0 | visual, hot needle | fibers/nd | 500–1300 | [127] |
| Ladoga Lake with tributaries | Atlantic | S | filtering device, 0.10 mm/DM | nd | visual, hot needle | nd/nd | 20–2400 | [124] |
| Neva River, mouth | Atlantic | S | filtering device, 0.10 mm/DM | nd | visual, FT-IR, Raman microspectroscopy | fiber/PET | 3000 | [125] |
| Lakes in Altai | internal watershed | S | direct sampling/DM | 5.00 | visual, SEM/EDS | fragment, film/nd | 11,000 | [75] |

Note: Abbreviations: S—spatial, S/T—spatio-temporal, nd—no data, DM—direct measurement, IM—instrumental measurement, C—calculation, FT-IR—Fourier-transform infrared spectroscopy, SEM/EDS—scanning electron microscopy combined with energy-dispersive X-ray spectroscopy, DSC—differential scanning calorimetry.

**Table 2.** Published data on the abundance of MPs in the bottom sediments of the Russian rivers and lakes (data are presented with increasing concentration).

| Water Body | Drainage Basin | Character of the Study | Detection and/or Identification Methods | Predominant Shape/Size Range/Polymer Types | Average MP Content, Items/kg Dry Weight | Reference |
|---|---|---|---|---|---|---|
| Malaya Neva River, mouth | Atlantic | S | visual, FT-IR, Raman microspectroscopy | fiber/PET | 30.0 | [125] |
| Smolenka River, mouth | Atlantic | S | visual, FT-IR, Raman microspectroscopy | fiber/PET | 60.0 | [125] |
| Neva River, mouth | Atlantic | S | visual, FT-IR, Raman microspectroscopy | fiber/PET | 120 | [125] |
| Ladoga Lake with tributaries | Atlantic | S | visual, hot needle | nd/nd | 60–2000 | [124] |
| Yenisei River | Arctic | S | visual, hot needle | fiber/nd | 353–489 | [123] |
| Nizhnaya Tunguska River (Yenisei River system) | Arctic | S | visual, hot needle | fiber/nd | 235–543 | [123] |
| Lake Onego | Arctic | S/T | visual, Raman spectroscopy | fiber/PC, PE, cellophane, PAN | 2189 | [74] |

Note: Abbreviations: S—spatial, S/T—spatio-temporal, FT-IR—Fourier-transform infrared spectroscopy. In the course of the first study of MP distribution in the Baikal waters with instrumental confirmation of the polymeric composition of particles, using nets with a mesh diameter of 0.30 mm for sampling [130], 0.27 items/m$^3$ of MPs was detected (Table 1). The sampling points were located in the zone near the southeastern shore of Lake Baikal and in one of the most visited places by tourists—the Small Sea Strait. According to the chemical composition, the particles were identified as PE (50%), PP (40%), and PS (10%).

The MP pollution of bottom sediments in the large Onego Lake was assessed [74]. These studies were conducted in the pelagic part of the lake, near the mouth of the Shuya River and the Petrozavodsk Bay. The MP content in the bottom sediments (upper 5 cm layer) varied from site to site and averaged at $2.189 \pm 1.164$ items per kilogram of dry bottom sediments (Table 2). The possible reasons for the relatively intense MP accumulation in the bottom of Onego Lake are both the proximity of MPs sources and the peculiarities of particle accumulation in the water body. In general, fibers were the most numerous type of particles compared to others, with their share being $61 \pm 13\%$ of the total detected MPs, followed by fragments, spheres, and films. Polycarbonate (PC), PE, cellophane, and polyacrylonitrile (PAN) were the most common types of polymers and together accounted for more than half of all detected MPs. It is noted that the distribution of particles largely depends on the physicochemical characteristics of bottom sediments, primarily their granulometric composition. The predominant accumulation of fibers is related to the medium silt fraction (0.01–0.05 mm) and indicates the presence of significant zones of MP accumulation in the lake.

In 2020, a large-scale survey of the entire length of the Volga River, including the upper, middle, and lower reaches, was conducted for the first time, in order to establish the MP distribution and identify their potential sources. The screening of 34 samples from the Volga surface waters, taken using a Manta sampler with a mesh diameter of 0.30 mm, showed that the MP content varied from 0.16 to 4.10 items/m$^3$, with an average content of 0.90 particles per cubic meter and an average mass of 0.21 mg/m$^3$ [126]. Using a differential scanning calorimetry, it was found that MPs in the Volga River were represented mainly by fragments of polyethylene and polypropylene of various shapes and colors, which reflects their different origin. Fragments, microfilms, and fibers in different proportions were found in the water samples. The authors of the study suggested household plastic waste being the main source of MPs entering the river. The role of large cities in river pollution was confirmed, since the highest concentrations of MPs were recorded in the water of the Volga River near and downstream of large cities, and the minimum concentrations were registered upstream the settlements.

This assumption was confirmed in another work. In the course of a comprehensive hydroecological study of the urban water bodies of the large Nizhny Novgorod city, associated with the Volga River, the MP content in water was screened [127]. It was found that the surface and underground waters of Nizhny Novgorod were heavily polluted with synthetic microfibers of anthropogenic origin: most of them were represented by brightly colored microfibers with a length of 90 to 2000 microns (Figure 2). The average particle concentrations were 0.50–1.30 particles per liter or 500–1300 items/m$^3$ (Table 1). The highest concentration of microfibers was observed at the mouth of a Volga tributary, the Levinka River.

The results of a 2019–2020 two-year monitoring of quantitative content of MPs at the mouth of the Northern Dvina have been published [84]. The Northern Dvina is one of the largest rivers in the European Arctic, flowing through populated areas with developed industry and flowing into the White Sea. This study was of a monitoring nature: samples were taken every month in the Korabelny arm of the river delta from September to November 2019 and from May to October 2020 by trawling with neuston nets with a mesh size of 0.33 mm. The volume of filtered water was calculated based on the length of the transect, the size of the nets, and the speed of the ship (Table 1), in contrast to the two previous works, where the volume was measured directly [125] or determined using a water meter [126]. A clear trend of seasonal variability in MP concentrations was not revealed. The actual MP concentrations fluctuated within 0.003–0.01 items/m$^3$ or 0.02–0.04 mg/m$^3$. The chemical composition of plastic particles in the Northern Dvina was determined using a FT-IR spectrometry. Most of the detected MPs were represented by 52.6% of polyethylene, followed by 36.8% of polypropylene. Fragments prevailed among the particles (82% of the total analyzed MPs). The average content of particles in the waters of the mouth of the Northern Dvina (0.007 items/m$^3$) was very low compared

to other available data on the fresh surface waters of the Russian Federation (Table 1); however, it was comparable to the data on the MP content in the waters of the Barents Sea (0.005 items/m$^3$) [22]. It might be due to the peculiarities of particle redistribution in the river system. Interestingly, the average mass concentration of MPs at the mouth of the Northern Dvina was 18.5 µg/m$^3$, which exceeded those for the Barents Sea and the Arctic Ocean (12.5 µg/m$^3$ and 3.70 µg/m$^3$, respectively) [22]. These data may indicate that the Northern Dvina is one of the significant sources of plastic pollution in the ocean, along with other rivers flowing into the Arctic seas.

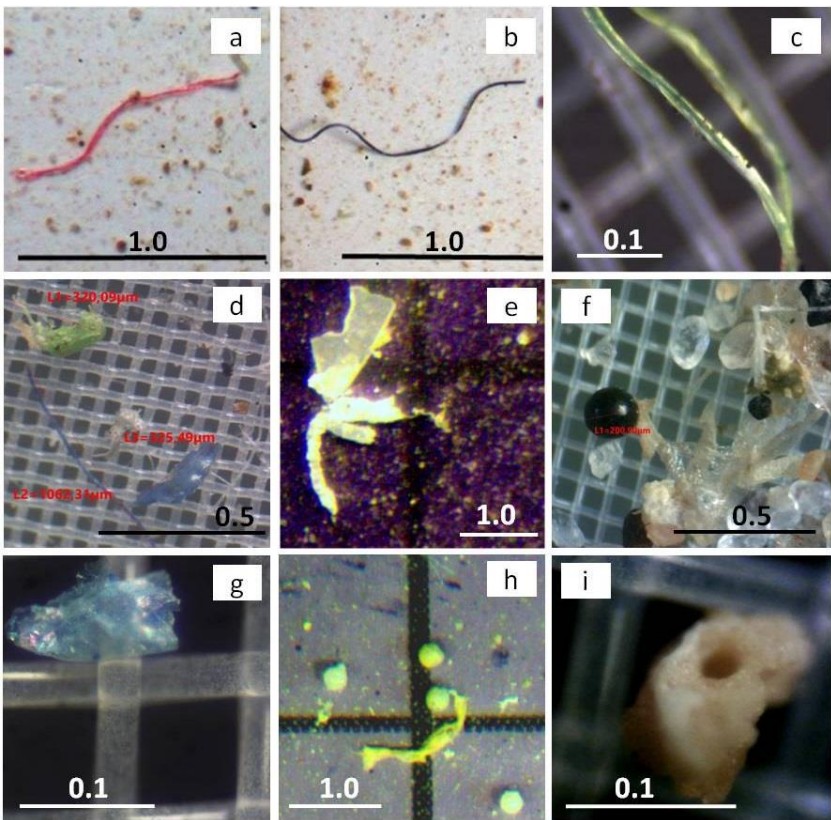

**Figure 2.** Microphotographs of MP particles from Russian freshwaters: fibers from the water and bottom sediments of the Yenisei River (**a,b**) and urban river in Nizhny Novgorod (**c**) [124,127]; fragments, films, and fibers from the urban river in Nizhny Novgorod (Ershova et al., unpublished); fragments from the water of the Ob River fragments and spheres from the urban river in Nizhny Novgorod (**d,f**) (Ershova et al., unpublished); (**e**) (Frank et al., unpublished); film from the water of Neva Bay (**g**) (Ershova et al., unpublished); spheres and film from the water of the Tom River (**h**) (Frank et al., unpublished); and foam from the water of Neva Bay (**i**) (Ershova et al., unpublished). Scale bars are in mm.

Another project to survey freshwater ecosystems associated with the Arctic seas of the European part of the Russian Federation was implemented in 2020 in the White Sea basin [34]. To obtain objective data, 500 L of water was filtered in each site. In the water samples from the Kedovka, the Koida, the Onego, and the Vaga Rivers, a range from zero to six MP particles per cubic meter was registered, with 1.67 items/m$^3$ being the average (Table 1). The maximum content of MPs was found in the Vaga River, a tributary of the Northern Dvina River. FT-IR spectrometry showed that, in most cases, the particles were represented by PE and PVC. The flow of plastics into the water bodies in this region could originate from unauthorized dumps, landfills, and domestic wastewater. The MPs in the waters of the floodplain lakes were distributed very unevenly, with the average content being 0.50 items/m$^3$, while no MP particles were found in three of the four sampled lakes [34].

2.3.2. Microplastics in the Freshwater Bodies of the Asian Part of the Russian Federation

The study of MP pollution in the continental waters of Western and Eastern Siberia began at about the same time as the projects to assess the MPs in the waters of the European part. Of all water bodies in the Asian regions, Lake Baikal attracts most of the research attention (Figure 1) due to its status as the largest and oldest natural reservoir of fresh water included in the UNESCO World Heritage List [133]. Preliminary studies conducted at tourist sites in the southern part of the lake showed the pollution of the surface waters with MPs, mainly as fibers and fragments, at a concentration four times higher than that in a large lake in Mongolia, the Hovsgol Lake, that was surveyed in parallel [128]. In terms of cubic meter, the MP content in the Baikal waters was 1.07 items (Table 1). Unfortunately, this publication does not contain a detailed description of the sampling process, the mesh diameter of the nets, as well as the method for determining the volume of filtered water; apparently, a calculation method was used. In the littoral zone of South Baikal, in nine representative water samples of 16–52 $m^3$, a range from 0.51 to 6.46 items/$m^3$ of MPs was recorded, with more than half of the particles (55–63%) being microfibers [129]. Fragments were much rarer and single granules were observed. A recent publication [132] presents a dataset on sewage pollution, periphyton, benthic macroinvertebrate community, and food web structure of Lake Baikal and includes data on the MP content in the waters along the southern shore of Lake Baikal at a depth of about 75 cm. According to the screening data, the average MP content in the waters at 17 points was 1.79 ± 1.34 items/$m^3$ (Table 1). Since these studies were preliminary in nature, the polymer composition of the particles was not determined.

Other authors reported relatively high MP concentrations in the surface waters along the western shore of Lake Baikal, with an average of 291 ± 252 items per cubic meter of water and the range being 34–707 items/$m^3$ [131]. However, in this study, the majority of the particles was represented by so called "mini-microplastics" of <0.33 mm. Such particles accounted for 88% of the total MPs detected by the authors. The use of direct water filtration through a plankton net with a mesh diameter of 0.02 mm at the sampling site helped to detect the smallest particles. MP fraction of <0.33 mm is underestimated in most studies that use neuston nets and Manta samplers with the standard mesh diameter, which is reflected in the quantitative results. The polymer composition of the particles found in the Lake Baikal waters was determined using a micro FT-IR, which showed the predominance of PP (65%), followed by PET (16%), PE, PVC, and alkyd resin (4% each) and other polymers (7%).

A new study was devoted to the preliminary assessment of the content and distribution of MPs in the Lake Baikal ice, which covers the water area in winter [134]. Ice samples were taken from the southern part of the lake in the Bolshiye Koty Bay and analyzed for particle content. Fibers prevailed among the particles found in the Baikal ice; their content, determined visually, varied from 55.5 to 65 items per liter (55,500 to 65,000 items/$m^3$), with predominant fibers of 0.7–1 mm. Since only a visual examination with a light microscopy was used for particle analysis, the results could only be considered as preliminary data. It is known that the error rate during the visual analysis of particles without the verification of the polymer composition using physicochemical methods can vary from 20 to 70% [135,136].

Obbard et al. [137] and Peeken et al. [138] found MPs in sea ice cores. Both studies concluded that the concentration of MP particles in sea ice is significantly higher than that in the surrounding water, and, thus, it can serve as a source of secondary pollution during ice melting. Therefore, sea ice can act as a temporary storage for MPs. Estimating the MP content in surface ice can be a useful monitoring tool, since an increase in the particle concentration can affect the albedo and, as a result, disrupt normal ice melting [139]. When the surface of a water body is covered with ice, MPs tend to concentrate at the ice–water interface, which leads to the accumulation of particles at the underside of the ice cover [128]. The nature of the plastic pollution of natural ice and its physical and chemical bases are currently under study [140].

The following sources and causes of MP pollution of Lake Baikal can be named: (1) mass tourism and an increase in the amount of plastic waste along the lake shores [128–131]; (2) human activities and the lack of wastewater treatment systems in settlements along the lake shores [129,134]—local pollution from the sewage of the settlements along the shores is characteristic for Lake Baikal [132]; and (3) the Selenga River, which flows into the Baikal, can be a major source of MP pollution since its basin covers a vast territory, which includes such large cities as Ulan-Ude (Republic of Buryatia, RF), Ulaanbaatar (Mongolia), and many other settlements [131,134].

Similar factors and MP sources can contribute to the water pollution of the plain and mountain lakes of Altai (Southern Siberia). The water samples taken from the lakes at a depth of 30 cm revealed a high content of MPs, ranging from 4000 to 26,000 particles in a cubic meter of water [75]. The maximum was found in the mountain lake Dzhulukul, and the minimum was found in the plain lake Zludyri. Interestingly, in the water of the large, mineralized Lake Kuchuk, the MP content was not the highest, although one could expect an increased content of polymer particles due to high mineralization. On average, for six studied lakes, including the large ultra-fresh Lake Teletskoye in the system of the Ob River, $11,000 \pm 7000$ particles/m$^3$ were detected. There are reports of a comparable high MP content in the waters of China's lakes. For example, in the water samples from Taihu Lake, MP content varied from 3.40 to 25.8 items/L or 3400–25,800 per cubic meter of water [141], and up to 34,000 particles per cubic meter of water was detected in Poyang [142]. In comparison to the data on the surface waters in the Russian Federation, these values are very high. It is possible that the MP concentrations in the waters of Altai lakes were overestimated, since the volume of the analyzed samples was small (Table 1). In addition, the authors used SEM/EDS to analyze the particles, judging the belonging of the particles by their surface features and elemental composition, which does not guarantee accurate verification of their polymeric origin.

For a preliminary assessment of the pollution of the Siberian rivers, a study was conducted in 2020–2021 in the Ob and Yenisei River Systems flowing into the Kara Sea [120,123]. Surface water sampling in both river systems was performed using a Manta sampler with a mesh diameter of 0.33 mm (Table 1). However, in the case of the Ob, the volume of filtered water was determined by calculation, based on the flow velocity, the size of the sampler section, and the exposure time, whereas in the case of the Yenisei, a counter was installed on the sampler, which, in our opinion, contributed to a more accurate determination of the volume of the samples.

The quantitative content was determined, and the morphology of MPs was analyzed in the surface waters of the Ob River and its major tributary, the Tom River. The average number of particles in the surface waters varied from $44.2 \pm 13.0$ to $51.2 \pm 36.5$ items/m$^3$ and from 79.4 to 87.5 µg/m$^3$ in the Tom and Ob, respectively [120]. In general, microfragments were the most common particle type in the studied areas of these two rivers. The average content of MPs in the surface waters and bottom sediments of the Yenisei River and its remote and longest tributary, the Nizhnyaya Tunguska River, were estimated in 2021 [123]. The total content of particles with a diameter of 0.15 to 5.00 mm in the Yenisei water was $2.95 \pm 0.66$ items/m$^3$ (Table 1). In the surface water layer of the Nizhnyaya Tunguska River, an average of $2.58 \pm 1.87$ items/m$^3$ was registered, with a tendency to increase down the stream ($p < 0.05$). The average concentrations of MPs in the bottom sediments of the Yenisei and the Nizhnyaya Tunguska were $353 \pm 153$ and $422 \pm 241$ items per kilogram of dry bottom sediments (Table 2). Among the plastic particles extracted from the surface waters and the bottom sediments of the N. Tunguska and Yenisei rivers, microfibers, microfragments, and microfilms were found. In the bottom sediments of the Yenisei and, especially, the Nizhnyaya Tunguska, fibers clearly predominated (Figure 2). As in the case of Lake Onego [74], a relationship was established between the MP concentration in the bottom sediments and their physicochemical features and the formation of microfiber accumulation zones. The average MP content in the bottom sediments sampled at different sites of the Yenisei River System depended on their total organic matter content (r = 0.952).

The possible sources of MPs in the rivers of Siberia include improper disposal of plastic waste in human settlements and diffuse accumulations of plastic waste along riverbanks; wastewater treatment plants (WWTPs) are the most likely sources of particle entry into the surface waters of Siberian rivers near large cities. In the northern and sparsely populated areas where the rivers Ob and Yenisei and their tributaries flow, fishing can also be a significant source of microfibers. Sport fishing as a specific source has previously been identified in the case of the relatively clean Dalälven River in Sweden, which flows through an area with less than 250,000 inhabitants [83]. In the northern part of Western and Eastern Siberia, fishing is a traditional source of food for the local population; river fish is the most important component of their diet [143].

Thus, according to the data from the first few years of studies on MP pollution of the Russian Federation continental fresh waters, there is a huge spread in the contents of particles both in waters and in bottom sediments. The minimum registered average MP concentration in the waters was 0.007 items/m$^3$ at the mouth of the Northern Dvina, and the maximum was 11,000 items/m$^3$ in Altai lakes (Table 1). The same picture was observed for the data on the MP contents in the freshwater bottom sediments: the detected minimum was 30.0 items/kg at the mouth of the Malaya Neva River, and the maximum was 2189 in the sediments of Lake Onego (Table 2). Among the reasons of such differences are the hydrological features of the studied bodies, the regional features of the MP sources, the volumes of MP inflow into rivers and lakes, natural conditions, as well as the use of a diverse set of methods by researchers for sampling and particle detection; in addition, in some studies, an unrepresentative volume of samples was studied (Tables 1 and 2). The representative volume of the studied water samples and the factors determining the lower size limit of trapped particles, in our opinion, are critically important for the objective analysis of MP distribution in surface waters. At the stage of laboratory analysis, the methods for separating particles by density (especially in the case of bottom sediments) and the use of modern techniques for analyzing the particle chemical composition to verify their polymeric origin can both be of crucial importance.

Despite the colossal differences in the methodology of particle sampling and analysis, there are common features of MP pollution in these continental freshwater bodies. Among the MPs detected in the surface waters and the bottom sediments, PE, PP, and PET particles predominate (Table 1). The most common types of MPs in the surface freshwaters are fibers and fragments (Figure 2). Fibers prevail in the bottom sediments (Table 2). From a hydrodynamic viewpoint, fibers are the lightest and most mobile form of MPs [144]. This is consistent with worldwide data, according to which fibers smaller than 1 mm are the dominant group of MPs in freshwater bottom sediments [115].

The typical sources of pollution for rivers and lakes are domestic wastewater, which contains a huge amount of microfibers of synthetic textiles, and fishing tackle. Along with these two sources of freshwater pollution by MPs within the country, the significant role of plastic waste fragmentation along the shores of water bodies due to the imperfect management of household waste has also been indicated.

2.3.3. Actual Research Trajectories of Microplastic Pollution of Continental Waters in the Russian Federation and Other Countries

Until recently, the problem of pollution of the Russian Federation continental waters by MPs has not received sufficient scholarly attention. Only starting from 2020, peer-reviewed sources have begun to report the actual concentrations of MPs in the waters of rivers and lakes in the Russian territory. To date, we are aware of a little more than 10 works on the quantitative estimations of MPs in the surface freshwaters (Figure 3), which, of course, is extremely scarce for an area of about 17 million km$^2$ with several of the largest freshwater bodies in the world.

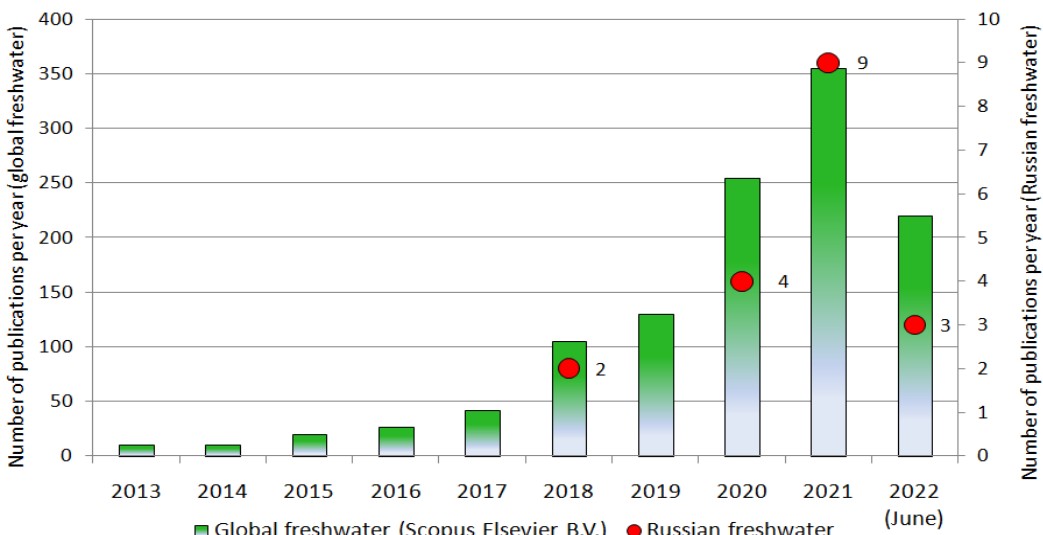

**Figure 3.** Publication dynamics on MP abundance in surface freshwaters over the past 10 years: number of publications around the globe (Scopus Elsevier B.V.) and in Russia.

Even less attention has been paid to the analysis of MP distribution in the bottom sediments (Table 2), despite the fact that they are the long-term pollutant reservoirs [107,145]. MP concentrations in the bottom sediments can be several orders of magnitude higher than those in the surface waters and the water column, which is reflected in the scale of particle uptake by benthic and benthivorous organisms and their further involvement in the food chains. Invertebrates that lead a bottom life make up to 90% of the food supply of fish [146], and bottom sediments accumulate not only MPs, but also various organic and inorganic pollutants [147]. Bioaccumulation of MPs in sediments can lead to the biomagnification of both plastics and associated toxicants [79].

In the near future, it is important for researchers of freshwater ecosystems specializing in MP pollution to focus on the following aspects:

The development of a unified methodology for sampling, processing, and analysis of MPs. Establishing standardized methods will allow more confident comparison of data from geographically dispersed areas. The results of quantitative analysis of MPs can be affected by the methodology used at every stage of the study, including the sampling method and the method of particle separation and identification. Our observations show that the accuracy in determining the volume of filtered water also affects the results of quantitative estimates. Key methodological challenges in quantifying MPs in surface waters are sampling using tools that provide a comparable minimum size of trapped particle [34,131] and the use of adequate physicochemical methods to verify the polymeric origin of particles.

More representative and regular studies need to be performed to obtain more accurate quantitative estimates. Most of the published works on quantitative assessments of MPs in freshwater ecosystems, not only in the Russian Federation but throughout the world, are a spatial screening that provides a "snapshot". In reality, the overall picture of pollution depends on many factors. To ensure representativeness, it is worth avoiding quantitative estimates of MPs based on the detection of particles in small volumes of water. The results obtained also depend on the location of sampling points in relation to large settlements and treatment facilities, and, to a large extent, they depend on the phases of the hydrological regime [60,148]. This once again confirms the need for systematic and space–time monitoring studies.

The transport and redistribution of MPs within the components of freshwater ecosystems needs to be studied, including waters, bottom sediments, shore soils, and biota. In addition to the need for detailed studies on rivers and lakes, a study is required to identify the role of dams and swamps in the processes of transport and accumulation of plastics by

surface water bodies on land [89]. Hydropower plants are becoming an important element in combating plastic pollution of water bodies. As their reservoirs accomplish the main regulation of river flow, their important additional function is to protect the upper and lower pools from floating debris. To this end, depending on the particular operating conditions of the hydropower plant, it is necessary to develop effective measures of controlling debris flows and removing floating debris from water [149].

Numerical modeling is one of the key tools for gaining insight into the distribution of marine debris, especially MPs. Integrating forecasting models of plastic flows and distributions is essential for supplementing the existing rare observations to estimate the number of MPs in different areas of the World Ocean [150,151]. Extending this approach by integrating simulation models and empirical observations can greatly improve understanding of the distribution and transformation of plastics, and especially MPs, in the marine environment. In the last decade, a series of numerical models have been developed specifically for the study of floating marine debris, on the basis of ocean hydrodynamic models of varying complexity [21,151–153]. Some of these models include the effects of currents, waves, and wind, as well as a number of processes that affect how particles interact with ocean currents, including their gravitational sinking/settlement, fragmentation, and degradation [154]. Living organisms can change the paths of MP movement in the marine environment by direct transfer and/or changes in particle density. The vertical transport of MPs initially positively rising to the surface can be significantly accelerated by the presence of aggregating algae [155]. In some models [153], MPs can be washed ashore, while in others, they can remain in the marine environment forever [151]. There has been research on applying numerical simulations to determine the source of plastic waste, i.e., solving the backward problem [156]. Ocean circulation models are also used to determine the most likely areas of oceanic accumulation. By linking simulation results with species distribution maps and other ecological information, it is possible to combine various data types to predict or identify risk hotspots for different geographical regions of interest [157,158]. Approaches also exist to identify the pathways or trajectories of plastic waste [158], identify hot spots, and develop scenario analysis tools to identify the potential sources and sinks of MPs. In addition to circulation models that assess the role of ocean currents and wave processes, other models can be used, such as risk models and bioaccumulation models (ecosystem-scale modeling) [154].

The consumption of MPs by living organisms and their transfer in freshwater food chains with extension to terrestrial food chains need to be investigated. Recent studies show that MPs are present in foods of animal origin that receive particles from the environment or in food chains, as in the case of aquatic organisms [159,160]. Many freshwater fish and invertebrates are caught and eaten by humans, which can be potentially hazardous to human health. The unresolved scientific issues requiring serious and large-scale studies include the following [161]: (1) the behavior of MPs in multilevel food webs, including their bioaccumulation mechanisms, and (2) the physiological effects and long-term consequences of exposure to MPs and associated pollutants in living organisms. In this regard, a comprehensive study of the pollution of freshwater environment components by particles of synthetic polymers and their trophic transfer, as well as a modeling of these processes, is extremely important for assessing the safety of the environment and the life quality of people in many regions.

If we try to single out the most "hot" topics for studying freshwater MPs in the Russian Federation, then there are two categories of continental freshwater bodies that, in our opinion, most of all need intensive and thorough research. These freshwater bodies are listed in the two points below.

Large oligotrophic freshwater lakes. Among the bodies that need constant pollution monitoring are large lakes that are significant as sources of fresh water and support for unique biodiversity: Lake Baikal (31,722 km$^2$) in East Siberia [133]; the lakes in the Northwestern part of the Russian Federation, including the Ladoga (18,300 km$^2$), the Onego (9800 km$^2$), and the Imandra (813 km$^2$) [162]; Teletskoe Lake (223 km$^2$) in the Altai Repub-

lic [163]. In addition, the importance of large lakes in the processes of MP accumulation in inland waters has been highlighted [74]. The progress of studying the pollution of Lake Baikal, Lake Onego, and other freshwater "pearls" is associated primarily with spatiotemporal studies and the organization of regular monitoring of the MP content in their waters and bottom sediments [74,128] and with the study of the interaction of MPs with living organisms and their behavior in food chains [129].

The largest rivers belonging to the Arctic basin. The rivers of the Arctic Basin require a more intensive and thorough study of the distribution and volumes of transferred MPs. Since plastic pollution has been detected in the Arctic Ocean [20,22,34,35,57], it is important to investigate the distribution of MP particles in the surface waters of rivers together with their accumulation in the bottom sediments and along the shoreline for an adequate assessment of the scale of potential pollutant carrying out. Modern models [81] do not take into account the data on plastic carrying out with the flow of northern rivers. Therefore, careful monitoring studies of MP distribution in the waters of rivers flowing into the Arctic seas will be the basis for more complete quantitative estimates of the global MP cycle. To date, only the data on the assessment of the flow of MPs into the Barents Sea have been obtained for the mouth of the Northern Dvina River [84]. The first published data on the Yenisei and Ob Rivers [120,123] are the results of pilot studies conducted prior to starting full-scale surveys, in order to establish preliminary data critical for the planning of further studies of the Great Siberian Rivers.

## 3. Conclusions

MPs are complex environmental pollutants that have a global distribution in the oceans and continental waters. Compared to the marine environment, the distribution, sources, and behavior of MPs in freshwater ecosystems are still not well-understood. In the Russian Federation, studies of the MP distribution in land surface waters and the identification of pollution sources are at the initial stage. More attention is being paid to MP screening in geographically diverse aquatic ecosystems without adequate monitoring. According to the available data on MP pollution of the Russian Federation continental fresh waters, the average MP contents vary greatly both in waters and in bottom sediments. The minimum average MP concentration in waters was found to be 0.007 items/$m^3$ at the mouth of the Northern Dvina, and the maximum was found to be 11,000 items/$m^3$ in Altai lakes. For freshwater bottom sediments, the registered minimum was found to be 30.0 items/kg at the mouth of the Malaya Neva River, and the maximum was found to be 2189 items/kg in the sediments of Lake Onego. In surface waters and bottom sediments, fibers and fragments made of PE, PP, and PET predominate.

The lack of standardized methodological approaches for the detection and quantification of MPs in aquatic ecosystems makes it difficult to adequately compare and interpret the data obtained by different scientific groups. To increase the relevance of future studies, it is important to solve a number of methodological issues related to water sampling and sample volume accounting.

The mechanisms of MP redistribution in land surface waters and freshwater sediments are also under study. The key to their deciphering may be the further identification of factors and forces influencing MP transport and redistribution between the components of ecosystems, using quantitative estimates based on spatiotemporal studies and modeling.

**Funding:** This research was supported by the Russian Science Foundation under the project No. 22-27-00720 "Abundance and accumulation of microplastics in Siberian Rivers".

**Institutional Review Board Statement:** Not applicable.

**Informed Consent Statement:** Not applicable.

**Data Availability Statement:** Not applicable.

**Conflicts of Interest:** The authors declare no conflict of interest.

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
