# Peer review of "Microplastics in Freshwater: A Focus on the Russian Inland Waters"

_water, doi:10.3390/w14233909_

Round 1
Reviewer 1 Report
The article is well written and is a significant study required to comprehend the situation of plastic pollution in the Russian aquatic systems. The authors have given their best to write the article and gathered a lot of information. This review can be one of the breakthroughs for the scientific community and can serve as a basis to interpret plastic accumulation in a big nation like Russia. The results can attract a large readership in the field.
1) The authors must check all the references for their appropriate citing. For. e.g. line 374, they cite Litterbase project [133] however, the cited paper does not describe the content. It is not possible to go through all the publications for a reviewer, hence, it is advised to recheck the citations.
2) In Fig. 1 The authors can improve it for more clarity showing the gradient of color for the intensity of plastic pollution. It is not clear what does 10-5, 10^3 and 10^10 represent. In India, it showed some points however, no supporting literature or description was found in the article.
On the other hand, for reference 133, the data was not highlighted in the figure. Please check.

Author Response
- The authors must check all the references for their appropriate citing. For. e.g. line 374, they cite Litterbase project [133] however, the cited paper does not describe the content. It is not possible to go through all the publications for a reviewer, hence, it is advised to recheck the citations.
Thank you for the comment. All the citations through the manuscript have been carefully checked and arranged in the correct order.
2) In Fig. 1 The authors can improve it for more clarity showing the gradient of color for the intensity of plastic pollution. It is not clear what does 10-5, 10^3 and 10^10 represent. In India, it showed some points however, no supporting literature or description was found in the article.
On the other hand, for reference 133, the data was not highlighted in the figure. Please check.
We are grateful for this comment. For Figure 1 we used the Global litter distribution map created as part of the LITTERBASE project (https://litterbase.awi.de/litter) as a background only by adding data on the Russian fresh waters. The diameter of the circles reflects the particle concentrations (the corresponding explanation was added to the Figure legend).
Reviewer 2 Report
The manuscript “Microplastics in freshwater: Focus on the Russian inland waters” reviews the data about the distribution of MPs in the freshwater ecosystems of Russia.
The paper is interesting and well organized. It can be published after minor revisions:
- All references are in a wrong order with a wrong citation number;
- There is some mistake in the page numbers;
- Line 245: why artificial polymers instead of syntetic polymers?
- Line 298-299: there is a repeated sentence;
- Table 1: pay attention to the column width (characte r);
- Some abbreviations are not explained (ex. Line 633 WWTPs);
- Line 647 and 812: why – 2,189 with sign - ?
Author Response
The paper is interesting and well organized. It can be published after minor revisions:
- All references are in a wrong order with a wrong citation number;
Thank you for the comment. All the citations through the manuscript have been carefully checked and arranged in the correct order.
- There is some mistake in the page numbers;
We are sorry about the wrong page numbering. It has probably appeared during the placing of our document into the journal template. We will make sure that the published version will have the right page numbers.
- Line 245: why artificial polymers instead of syntetic polymers?
Replaced by "synthetic polymers".
- Line 298-299: there is a repeated sentence;
Corrected. The repeated part was removed.
- Table 1: pay attention to the column width (characte r);
Corrected in Table 1 and Table 2.
- Some abbreviations are not explained (ex. Line 633 WWTPs);
WWTPs and FT-IR abbreviations are now explained in the text.
- Line 647 and 812: why – 2,189 with sign - ?
It was a dash, but we removed it in both cases so as not to confuse the reader.